# A Low-Cost Visible Light Positioning System for Indoor Positioning

**DOI:** 10.3390/s20185145

**Published:** 2020-09-09

**Authors:** Juan Carlos Torres, Aitor Montes, Sandra L. Mendoza, Pedro R. Fernández, Juan S. Betancourt, Lorena Escandell, Carlos I. del Valle, José Manuel Sánchez-Pena

**Affiliations:** 1Display and Photonic Applications Group, Electronic Technology Department, Carlos III University, Butarque 15, 28911 Leganés, Spain; aimontes@pa.uc3m.es (A.M.); smendoza@inst.uc3m.es (S.L.M.); pedrfern@ing.uc3m.es (P.R.F.); juansebastian.betancourt@uc3m.es (J.S.B.); lorena.escandell@uc3m.es (L.E.); jmpena@ing.uc3m.es (J.M.S.-P.); 2Optiva Media, Edif. Europa II, Calle Musgo 2, 28023 Madrid, Spain; carlosivan.delvalle@optivamedia.com

**Keywords:** indoor positioning system (IPS), LED beacon, smart systems, visible light communication, visible light positioning (VLP), VLC beacons

## Abstract

Currently, a high percentage of the world’s population lives in urban areas, and this proportion will increase in the coming decades. In this context, indoor positioning systems (IPSs) have been a topic of great interest for researchers. On the other hand, Visible Light Communication (VLC) systems have advantages over RF technologies; for instance, they do not need satellite signals or the absence of electromagnetic interference to achieve positioning. Nowadays, in the context of Indoor Positioning (IPS), Visible Light Positioning (VLP) systems have become a strong alternative to RF-based systems, allowing the reduction in costs and time to market. This paper shows a low cost VLP solution for indoor systems. This includes multiple programmable beacons and a receiver which can be plugged to a smartphone running a specific app. The position information will be quickly and securely available through the interchange between the receiver and any configurable LED-beacon which is strategically disposed in an area. The implementation is simple, inexpensive, and no direct communication with any data server is required.

## 1. Introduction

Currently, the technological evolution is carrying new concepts, such as the Internet of Things (IoT), where, positioning in indoor environments is essential for “real-time” tracking of people, devices, and tools [1]. Since the implementation of satellite navigation, GPS or Galileo on most mobile devices, outdoor positioning has become a must-have service in modern society. However, even with the wide range of satellite signals, they are not always strong enough to be used in indoor environments. Thus, the further development of a varied range of new technologies to meet the demand of indoor positioning systems is needed, including WLAN, BLE, ultra-wide band (UWB), ultrasonic wave, and Visible Light Positioning (VLP). Most of these solutions use radio technology; however, these solutions require a specific, complex, and expensive deployment [2,3]. Besides, radio technology can be compromised in specific deployment environments, including tunnel construction, mining, petrochemical plants and hospitals, among others.

However, the restrictions of radio-based solutions can be overcome with other types of systems, such as VLC systems, which can be deployed using the installed illumination infrastructure with small modifications. All these features generally provide the system a high density of light sources, that can be also easily limited by walls or other physical elements. In this way, the accuracy of a VLP system could be increased and would be less sensitive to the multipath effect. On the other hand, VLC systems are not affected by electromagnetic fields and do not create interference with other communication devices. However, VLP needs a direct Line of Sight (LOS) and the lights must stay on, even during daytime.

A VLP setup is basically composed of a transmitter and a receiver. The main components on the transmitter is a commercial off-the-shelf LED (or a LED array), which is controlled by a programmable driver. The transmitter can be named as “Beacon” because it has a fixed position and generally continuously sends the same pattern. The receiver incorporates a light detector or an image sensor to receive and decode the information received from the beacons.

In the VLP receiver, the position can be decoded using at least three different strategies: range-based methods, range-free methods and fingerprinting [3]. The most straightforward localization algorithm, but least accurate, is the range-free approach (also known as proximity or coverage approach) [4], where the location is determined with an identification code (ID) sent through the beacon and is associated to a known area. In this approach, a mobile target is located and positioned inside the area illuminated by the strongest beacon footprint.

The received ID is used to search the transmitter device record in an LUT or database and retrieve the location information of the transmitting device, which is then reported as the location of the mobile device [5,6,7]. The system precision can be improved using the received signal strength indicator (RSSI) algorithm [8,9,10], because the distance between the transmitter and receiver increases as the strength of the received signal decreases. Therefore, the receiver position can be estimated using the strengths of the received signals.

The effects of the optic power transmitted are unpredictable. When the signal decreases due to the distance between the transmitter and the receiver, the interference effect of obstacles or reflected waves are larger, limiting the precision of the localization method.

Another frequent source of interference is ambient light because it changes the strength of the received signal and, in outdoor or windowed environments, can produce unpredictable changes that effect positioning. Either way, the proximity algorithm has low precision since, in indoor positioning, a couple of meters can be the difference between being in one room or another.

Other techniques, such as trilateration based on RSSI [3,10,11,12] or fingerprinting [3,13,14,15,16,17], have been studied to improve system precision. However, these techniques have been discarded in this work. The literature presents other algorithms that use other light characteristics, such as the angle of arrival (AOA), the time of arrival (TOA) and the difference in time of arrival (TDOA). However, all these algorithms have drawbacks, including the complexity of the receiver circuit, the weight and volume of the deployed hardware, consumption, and the processing time of the implemented algorithm, making them not viable solutions to the indoor positioning problem.

In this paper, we propose a solution that works with the proximity algorithm, seeking to improve implementation problems in indoor positioning systems. Hardware and software can transform any mobile device, such as smartphones, tablets, or virtual/augmented reality glasses, into a VLC receiver device. Then, the device can obtain its position transparently, as how it could obtain its position from other geolocation systems. However, this solution is not based on a distributed system where energy consumption could be redistributed between different elements of the system, as proposed in [18,19], especially when multiple receivers are in play.

On the other hand, the design and development of a small receiver device that does not need external power and has low consumption are addressed. Furthermore, in this paper, we show the viability of a system using the designed low-cost receiver in combination with an Android or IOS device to localize and follow people or resources in indoor environments.

## 2. Materials and Methods

### 2.1. System Design and Protocol

The proposed system consists of multiple light-emitting diodes (LED) lamps that perform as beacons, sending a message indicating their position through the illumination beam. This occurs when a user carrying a mobile device (with a low-cost detector connected) is in the range of one of the light beams, as shown in Figure 1. Furthermore, each lamp could send additional information about near services and promotions, among others.

An important design restriction is not to overlap light cones emitted from adjacent beacons. The LED beacons were designed to illuminate a narrow area reducing the possibility that multiple beacons share the same area/channel (Figure 1).

In the scope of the application, two users were required: a standard user and admin (Figure 1). The former can only make use of a navigation app. The administrator uses another independent application that has advanced functionalities and that can modify the status and information of the beacons.

The info stored on the beacons can be modified using an independent infrared interface controlled by an Admin app. This app runs on a standard mobile device which incorporates an infrared interface, and a VLC receiver module, as with the rest of users.

The firmware running on the MCU is interrupted when a command is received from an IR interface. In that case, the device falls into a programming mode that prepares the beacon to receive the information to broadcast. This operation can be executed from a mobile device that runs a specific administration application that, among other functions, controls IR module to update the beacon info. A microcontroller runs a firmware that follows the flowchart represented on Figure 2. The MCU manages both the sending of the VLC frame and the reception of the IR signal to update the beacon.

The transmission protocol and the transmitter and receiver designs are discussed below.

#### 2.1.1. Transmission Protocol

The selected modulation to transmit the data is Frequency-Shift Keying (FSK), where each bit is coded in frequency, associating each logic level to a different carrier frequency. The resulting modulated signal is:(1)$$S_{FSK}\;=\;A\sin2\pi t\left(f_{l}\;+\;data\left(f_{h}\;-\;f_{l}\right)\right),\;data\;\to \left\{0,1\right\}$$
where A is the amplitude of the signal. The mark frequency (bits “1”) is a signal of *fl* = 1.2 kHz, and the space frequency (bits “0”) is a signal of *fh* = 2.2 kHz. Each symbol has a duration of Ts = 8.4 ms, as shown in Figure 3a. The transmission frequencies limit the maximum bit rate, but the fact that the user would not be in movement is enough to receive all the messages.

The first element in the message structure is a six-bit header, and the next three bits identify the lamp. Then, four bits know the package, and the last eight bits are the messages, as shown in Figure 3b.

The lamp transmits its geographic coordinates in sexagesimal format (degrees, minutes, and seconds). The information could be, for example:Latitude: 40° 18′ 00′’ NLongitude: 03° 43′ 00′’ W

Each character is encoded in eight bits and forms the message body. The lamp must send eight packets of information to receive all the geolocation information. These packets are divided into four for longitude and four for latitude. The packets are sent continuously in a cyclic way so that when the user is under one of the lamps, the packets receive their exact location.

#### 2.1.2. LED Beacon

The beacon is a programmable driver that is responsible for sending a data frame, continuously. Additionally, the beacon can modify the stored info when required. Therefore, it is necessary to add a microcontroller to format and send the information through the LED lamps and an IrDA interface to receive the info to be changed. The internal structure can be shown in Figure 4.

The beacon includes three sub-circuits: the IR interface, the Driver and the MCU. The IR Receiver permits the beacon configuration at 9600 bps and relative short distance using the Admin-App.

The LED bias current is approximately 42 mA, generating an illuminance of 88 and 24 lux at distances of 100 and 200 cm, respectively. The microcontroller used for this specific solution is an STM32L152 with a clock frequency running at 32 MHz, because of its flexibility in use, small size, and fast software configuration. A 2N6029 BJT transistor, is used to drive the LED. To bring power for the emitter systems, an IC LSD05-15B05S is used. It is configured with a few components to stabilize a 5 V output voltage, generating enough current for the LED and the microcontroller. To generate the mark and space signals, the internal oscillator of the microcontroller is used as a clock, creating a PWM output in the desired pin. The bit time is controlled with a separated timer to ensure that the time maintains the same, even for “1” and “0” bit frequencies.

#### 2.1.3. Optical Detector and Conditioning Electronics

A receiver is coupled to an Android device through the audio jack to capture the VLC signal. The switched light sent by the beacon is captured by this optical receiver, which amplifies and filters the signal to be processed using a user application. The optical detector consists of a photodetector and an amplification circuit. Figure 5 shows the block diagram of the receiver module. Note that energy supply for the amplifier circuit is provided by the Android device through the audio jack plug, avoiding the use of any external power module.

The receiver schematics and views of the implemented module are shown in Figure 6. A BPW21R photodetector (D1), which was especially designed for high precision linear applications, is used. It has a spectral bandwidth range from 420 nm to 675 nm with a maximum sensitivity peak at 565 nm. The current generated by the photodetector is used to bias a BJT transistor 2n2222 (Q1). A 1 µF capacitor (C1) is used to eliminate the DC component and low noises.

The circuit small signal behaviour was analysed to obtain the required voltage output. At first, the small signal equivalent circuit (Figure 7a) is divided into the input and output nets knowing the intrinsic transistor values (*r_π_* = 2273 Ω, *β* = 110). Then, with values of $R_{1}=154\;kΩ,{\;R}_{load}=2\;kΩ\;$ the resulting gain is $\sim 10^{5}\;\left[V/A\right]$ (Figure 7b).

Therefore, the receiver device has a high gain, as well as input and output impedances with intermediate values. These characteristics make it ideal for working in intermediate stages.

#### 2.1.4. Signal Processing

After the signal is collected and amplified by the receiver device, it is sent to the Android device through the audio jack to be processed. A custom Android application was developed to collect, process, and show the information to the user using the integrated development environment (IDE), Android Studio 3.1.1 (JetBrains, Prague, Czech Republic; Google, Menlo Park, CA, USA). The received signal passes through the following multiple stages: reception, signal reshape and storage, signal conditioning, demodulation, and synchronization, as shown in Figure 8.

The sampling frequency required for the audio AD converter in the receiver is estimated using *fh* as it has a greater number of cycles (*n*) per symbol. To obtain the bit rate of a signal with frequency *fh* in *n* cycles,
(2)$$f_{s}\;\geq \;k\cdot BW,$$
(3)$$bit\;rate\;=\;\;1/T_{S}\;=\;BW/N\;=\;f_{S}/k\cdot N$$
where *k* must be at least two.

As the duration of each bit is 8.4 ms, and the bit rate is 119 bps, the A/D converter sample frequency configured is 32,000 samples per second (sps).

The necessary samples per bit can be obtained as follows:$$\frac{32,000\;\mathrm{sps}}{119\;\mathrm{bps}}\;\approx \;269\;\mathrm{samples}\;\mathrm{per}\;\mathrm{bit}$$

To ensure that all the bits of the package are obtained, the input buffer must have a size of at least 42 bits. Therefore, the buffer size must be at least 11,298 positions (269 sps × 21 bits ×2 = 11,298 samples).

The captured signal is stored in the buffer in a pulse code modulation (PCM) format, because, as it is collected through the jack input of the Android device, the signal is treated as an audio signal. The recorded signals oscillate between positive and negative values.

The signal amplitude varies depending on the distance between the LED beacon and the receiver. As the modulation works over frequency modulation and not over amplitude modulation, the variances in the amplitude and the additive noise do not affect the symbol detection.

The signal amplitude varies depending on the distance between the LED beacon and the receiver module. As the scheme works using frequency modulation, any amplitude variation in the received signal does not affect the symbol detection.

If the optical receiver module lacks a flat response for all frequencies, the received signal amplitude may vary between the mark and space signals. Additionally, the low-pass filter used in the receiver driver will affect the amplitude. This situation is shown in Figure 9.

After digitalizing the signal, the demodulation process will be executed in the digital domain. Even when an FSK modulated signal can be demodulated in different ways, the one used in this system measures the signal period. Initially, the stored signal is reshaped using a simple threshold algorithm (Figure 10).

Then, the detection will be executed using a correlator algorithm, such as that described in [20] (Figure 11). The main idea in [18] is to find the optimal delay (*θ*), which causes a significant correlation difference between the two possible frequencies of the input signal (*fl* and *fh*).

The correlator output is determined as follows:(4)M(t) = cos(2πft)·cos(2πf(t + θ))
(5)M(t) = 1/2[cos(−2πfθ)−cos(2πf(2t + θ))

The second term is twice the input frequency and can be easily rejected using a low pass filter. However, the first term tends to be a DC component depending on the delay *θ* and the current input frequency. This component is just the sent data. Thus, choosing an optimal delay, the original data are recovered. To obtain the optimal delay value *θ*, that causes the maximum variation, Equation (5) must be maximized.
(6)$$Diff\;=\;\max(\cos\left(2\pi f_{l}\theta \right)\;-\;\cos\left(2\pi f_{h}\theta \right))$$

The previous analyses can be extrapolated to square signals. In our case, we applied a delay of approximately 429 µs (~14 bits of delay in the buffer) because our mark and space signal frequencies are 1.2 kHz and 2.2 kHz, respectively. Notice that the computational cost is decreased with respect to [20] using the detector algorithm as well as an XOR function instead of a multiplier. This simplification is possible because the signals involved in the correlation operation are previously shaped squares.

After applying the XOR function (see Figure 12), the resulting signal is passed through a digital low-pass filter to recover the original data.

After filtering, the input signals with a frequency of 1.2 kHz (“1” bits) result in a high output level, and the input signals with a frequency of 2.2 kHz (“0” bits) result in a low output level. An example of the filter output is shown in Figure 13.

The next step is to transform the obtained high values in ones and the low values in zeros. This step transforms the filter output into a square signal containing the message transmitted (Figure 14).

Once the input signal is demodulated, it is necessary to locate, in the buffer, the synchronism bits indicating the beginning of a frame. When the synchronization trace is detected, the algorithm can recover the information from that point (Figure 15).

The message begins with a header consisting of 6 synchronism bits, 5 high level bits ‘1′, and 1 low level bit ‘0′. In this way, when the receiver detects the header, there must be a specific number of bits indicating the lamp identifier, the corresponding number of packets of information, and the location coordinates.

## 3. Results and Discussion

The versatility of this system allows it to be installed on any Android device that has an audio jack port. In the laboratory, some tests were carried out with smart glasses, EPSON BT 200 model H560A (SEIKO EPSON, Suwa, Nagano, Japan), which also use an Android operating system, as shown in Figure 16. This system works perfectly with other types of devices without making any additional modification in both the software and hardware of the smart glasses.

The setup used to test the whole system is shown in Figure 17. Four beacons were used and the Android device with the detector attached was passed through the light beams (beam angle of 11 degrees) to verify the performance of the system and the Android application.

The maximum distance achieved between the receiver (Android device + photodetector) and emitter (LED lamp) was 200 cm. As it is the maximum vertical distance, the device was moved in a horizontal way to obtain the maximum horizontal distance at the maximum vertical distance, obtaining 75 cm as the maximum value that maintains a BER less than 10^−9^. As the position information can be sent in one data frame with a size of 14 bits, the minimum time that the user must be under the lamp is 117 ms. That minimum time is traduced in that the users can be moving at a maximum speed of 12 m/s to use our system and ensure data integrity.

In addition, as a part of the system, an indoor navigation application was developed, as shown in Figure 18. The application embeds a map where the user is located with the received location information from the beacon, as well as the lamp identification number.

The receiver was designed to reject possible interferences such as sunlight or artificial lighting. The signal amplitude in the receiver will variate depending on the distance between the LED beacon and the receiver. Multiple tests have been carried out in different hours of the day and other tests were performed, having the lights of the room switched on and off, and similar results were obtained with just a few differences.

We have not carried out a deep analysis of the energy consumption of the algorithm. It could be a very interesting task, and we will consider it in future works. Nevertheless, the application uses processing similar to standard audio recording applications, except for the FSK demodulation algorithm. Our first app version (running on android 7.0), uses 1.7% of the system consumption (verifying system records). Today, any mobile device can handle these requirements for hours, so we did not present it, and the work was more focused on the hardware implementation.

## 4. Conclusions

This paper proposes the design of a low cost and low power consumption VLC system as a solution to spread the use of a VLC-based indoor positioning systems in society, allowing any smart device to become a VLC receiver.

In the tests carried out, no positioning errors were detected while positioned in the coverage cone at the maximum vertical distance from the emitter. The device was tested with multiple LED lamps and different smart devices, such as augmented reality glasses, tablets and smartphones, among others. The system allows the user, with a low illumination level (~24 lux), to establish the device position with a precision less than 75 cm at 200 cm of the vertical distance between the emitter lamp and the receiver.

The developed application allows for indoor navigation, showing the position received from the lamp in a map. The obtained results also show that the system can properly work with users moving at a speed of up to 12 m/s.

The system described here does not need to have a database or a preloaded LUT to retrieve the location information because the message frame itself contains the coordinates of each beacon.

The sampling frequency required for the audio AD converter in the receiver is estimated using *fh* as it has a higher number of cycles (*n*) per symbol. The maximum bit rate obtained was 441 bps when using a sampling rate of 44.1 kHz and *n* = 10.

## Figures and Tables

**Figure 1 sensors-20-05145-f001:**
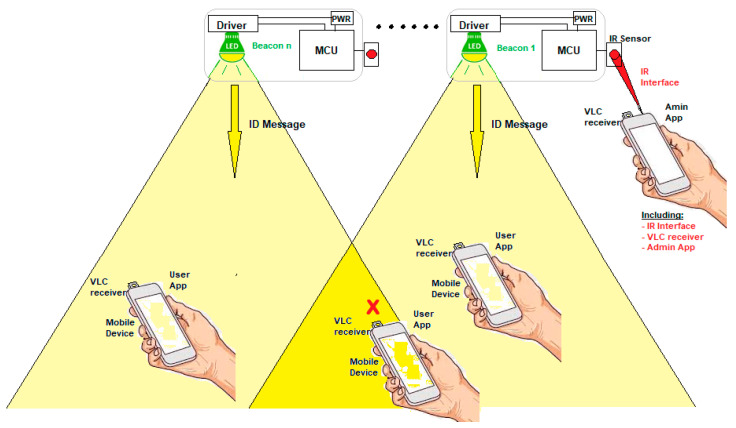
VLC beacon system based on proximity algorithm.

**Figure 2 sensors-20-05145-f002:**
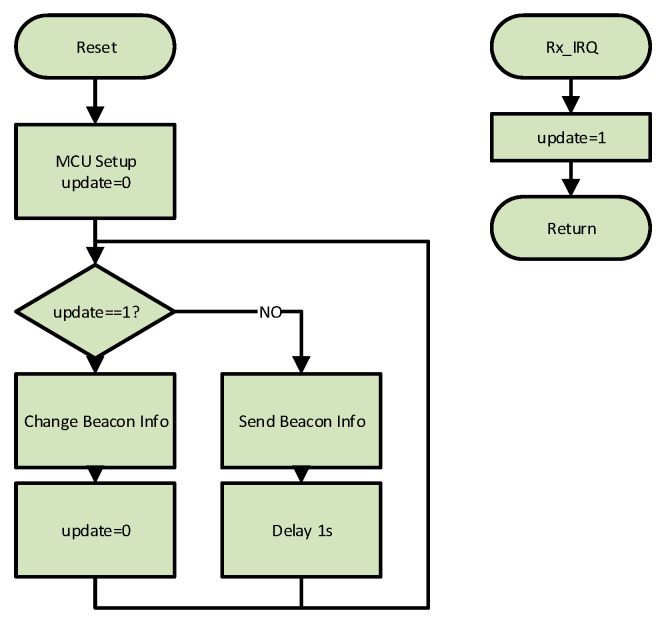
Flowchart of the beacon firmware.

**Figure 3 sensors-20-05145-f003:**
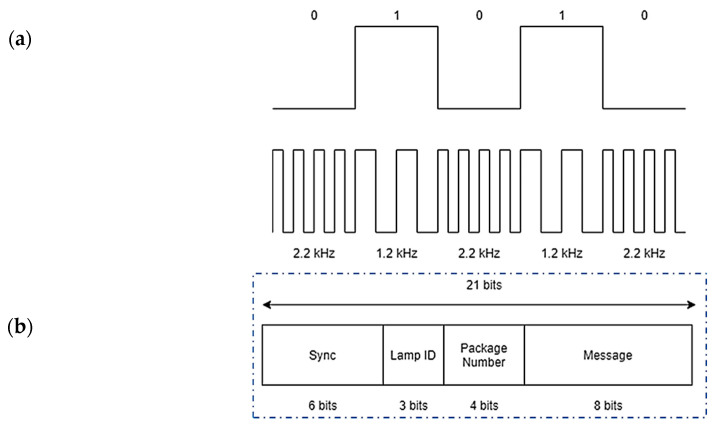
Message codification: (**a**) FSK modulation, (**b**) communication protocol.

**Figure 4 sensors-20-05145-f004:**
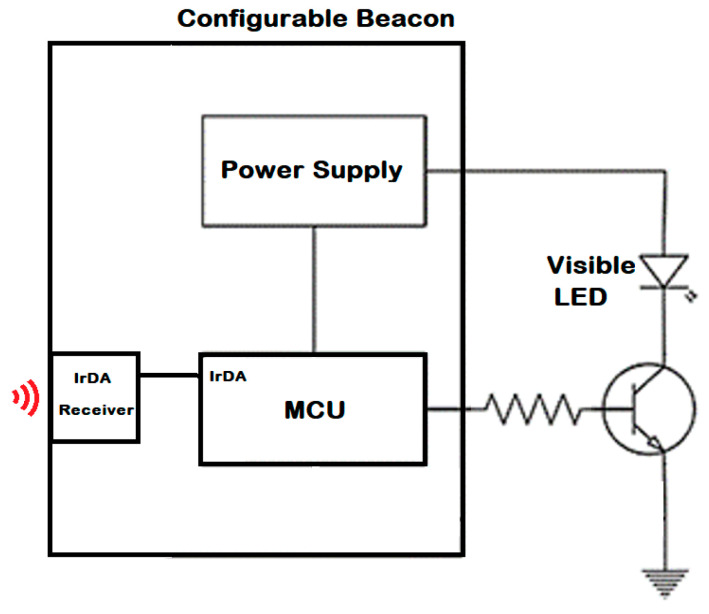
The beacon block diagram.

**Figure 5 sensors-20-05145-f005:**
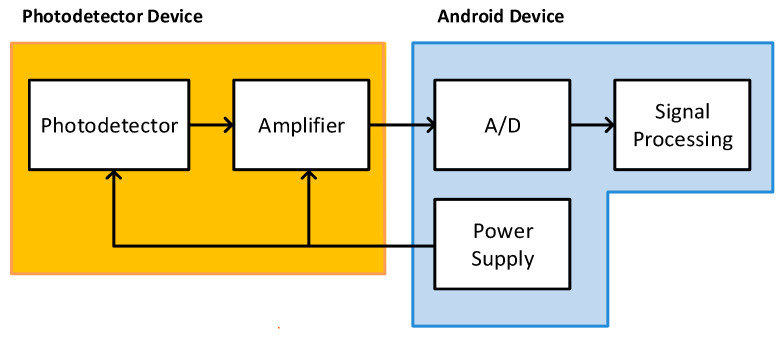
Block diagram of the receiver module.

**Figure 6 sensors-20-05145-f006:**
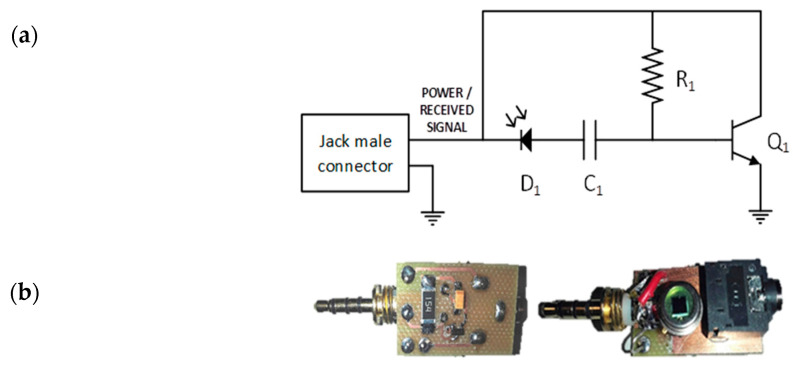
Receiver module (**a**) schematic and (**b**) PCB-Mount circuit.

**Figure 7 sensors-20-05145-f007:**
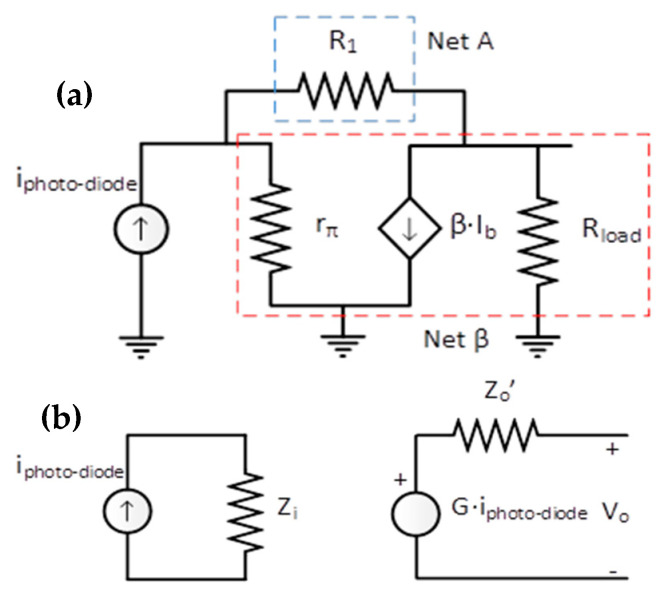
Receiver (**a**) small signal equivalent circuit and (**b**) simplified equivalent circuit.

**Figure 8 sensors-20-05145-f008:**
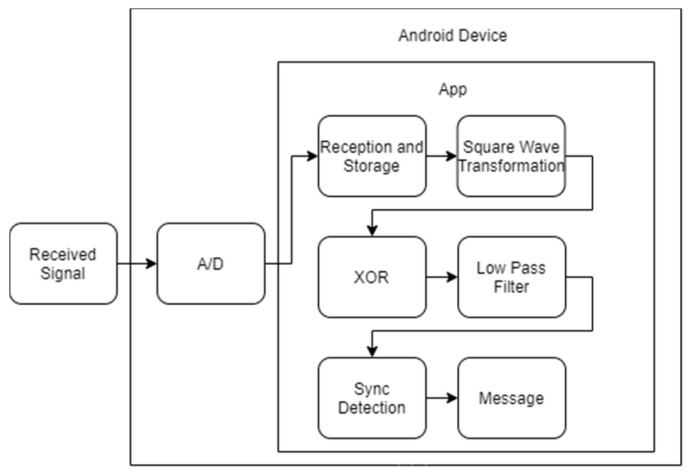
Application block diagram.

**Figure 9 sensors-20-05145-f009:**
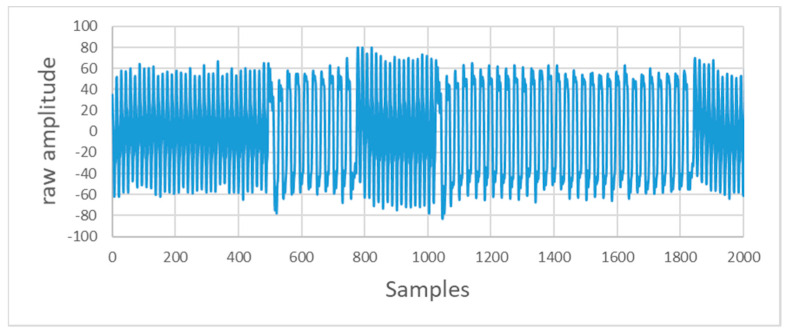
Signal recorded through the audio jack.

**Figure 10 sensors-20-05145-f010:**
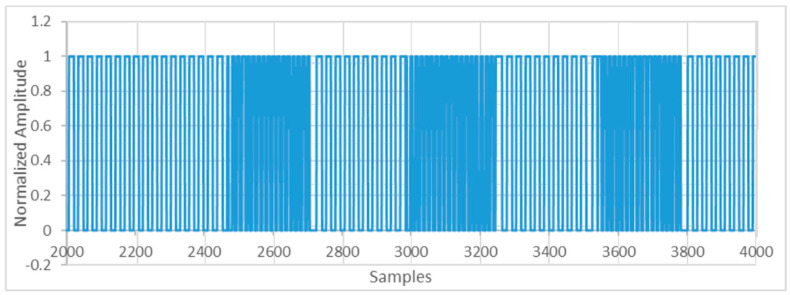
Reshaped signal through simple threshold algorithm.

**Figure 11 sensors-20-05145-f011:**
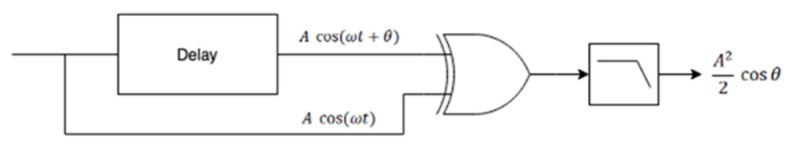
FSK Detector block diagram.

**Figure 12 sensors-20-05145-f012:**
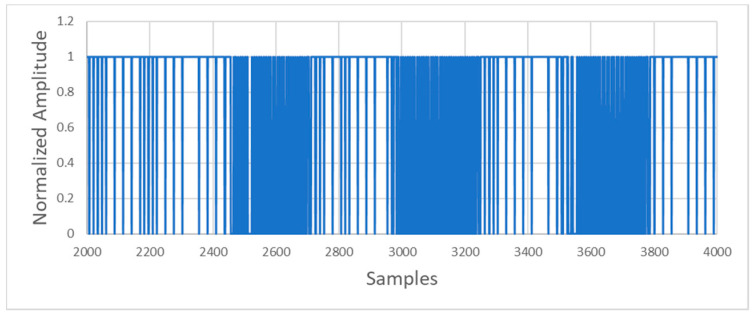
Signal obtained at the XOR output.

**Figure 13 sensors-20-05145-f013:**
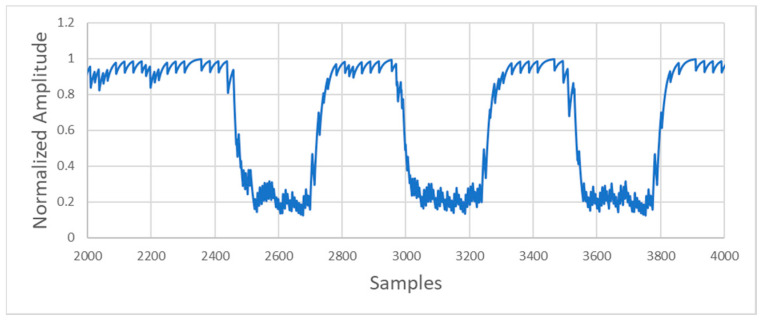
Detector output.

**Figure 14 sensors-20-05145-f014:**
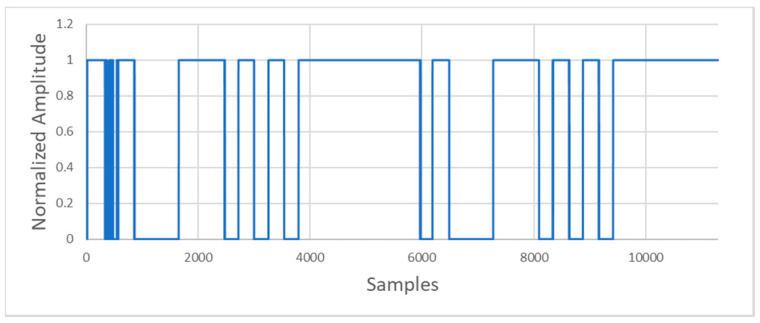
Signal obtained after the processing.

**Figure 15 sensors-20-05145-f015:**
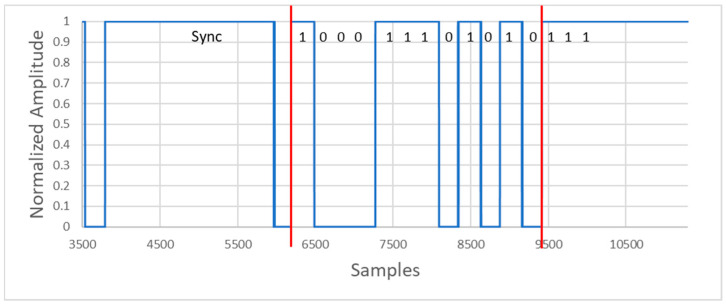
Synchronization and message bits after processing.

**Figure 16 sensors-20-05145-f016:**
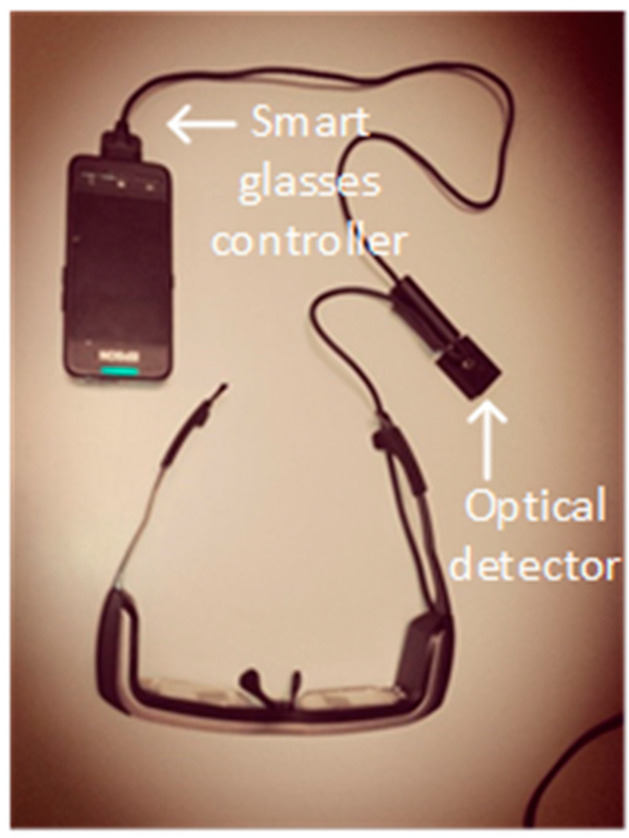
EPSON BT-200 smart glasses, model H560A, with receiver connected to the jack port.

**Figure 17 sensors-20-05145-f017:**
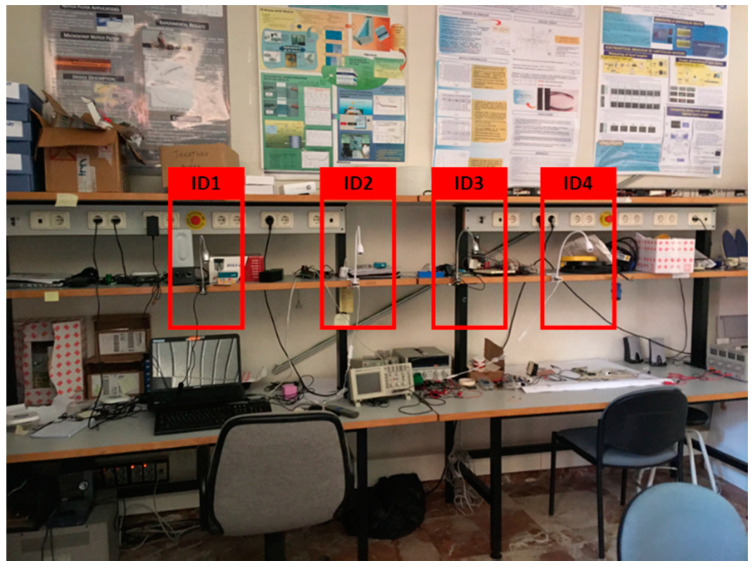
Experimental setup.

**Figure 18 sensors-20-05145-f018:**
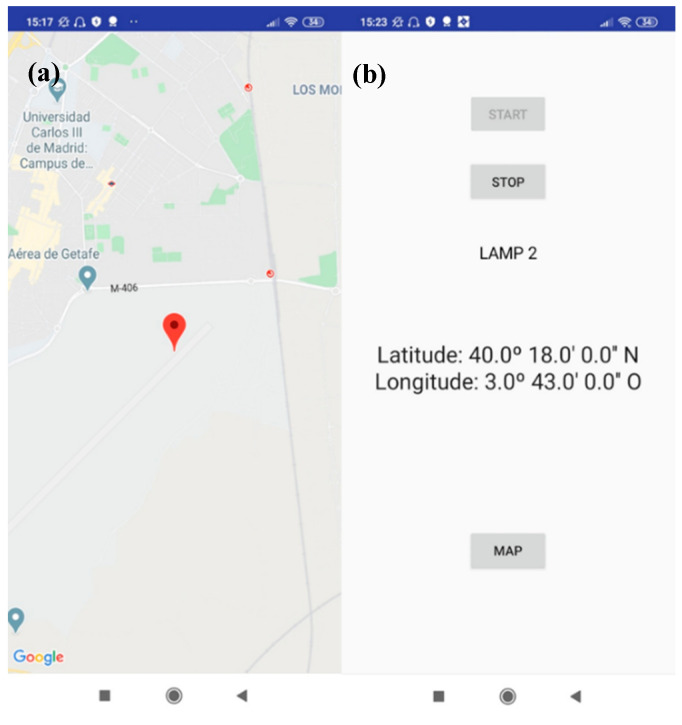
Application (**a**) map view of the obtained position and (**b**) obtained position in DMS format.

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
