# Peer review of "A Low-Cost Visible Light Positioning System for Indoor Positioning"

_sensors, 2020, doi:10.3390/s20185145_

Round 1
Reviewer 1 Report
This paper tackles the Visible Light Communication (VLC) problem which I think it could be thought of the next generation in indoor positioning.
The paper is surely interesting and the concept are clearly explained. However there are some points that must be improved before this manuscript could be considered ready for publication.
What happen if the user is far (but not too far) from the emitting LED ? Can some errors in the processing stage occur that can lead to positioning detriment? Authors should consider this.
What about if there is an occlusion during the transmission ? Can the algorithm recover the message correctly ?
Energy is also a crucial parameter if your algorithm runs over a smartphone. Can authors provide some hints about the energy request of your solution ?
Finally, taking all these observations into account, I suggest to add these references to strengthen your bibliography.
- “Smart Probabilistic Fingerprinting for Indoor Localization over Fog Computing Platforms” , 5th IEEE International Conference on Cloud Networking, 3-5 October 2016, Pisa, Italy, DOI:10.1109/CloudNet.2016.43.
- "Energy-Efficient Indoor Localization WiFi-Fingerprint System: An Experimental Study," in IEEE Access, vol. 7, pp. 162664-162682, 2019, doi: 10.1109/ACCESS.2019.2952221.
Reviewer 2 Report
1) The paper is an engineering application with apparently not much research innovation. I would try to make the research content, in terms of novel contribution and advancement of the state of the art, more clear.
2) how interference from neighboring lamps is addressed? The light cones will overlap to some extent (as visibile in Fig. 1), so how it is managed the multiple access to the medium?
3) I was wondering why the synchronization header is not directly used for detection and synchronization, through a matched filter. Is this possible in your case, and might it improve the performance?
4) It is not clear how the position is estimated. It seems that the algorithm is more a ranging than a full localization system, but then how the position error is computed? This part requires more clarification, and in general it would be better to see some curves for the performance assessment as a function of some parameter (typical ones are SNR, distance, BER)
5) English must be improved, some terms are also wrongly translated in technical jargon (e.g., packages instead of packets, timer instead of clock (where appropriate), convertor instead of converter, etc.)
Round 2
Reviewer 1 Report
I think that the authors have addressed all my concerns. They provide an accurate answer to all my raised points.
I suggest, as a future work, to think to include also a correction code within the transmitted message. This should improve the transmission and guarantee the correctness of the received data.
Finally, in my opinion, this paper can be considered ready for publication.
Reviewer 2 Report
the paper has slightly improved, but still remains a limited contribution; on the other hand, due to the difficulties of making experiments in this Covid time, and the interest in practical engineering solutions for this emerging topic of VLC, the paper could be interesting for the community.